# OpenReview forum: "Agentic Confidence Calibration"
_ICLR.cc/2026/Conference — ICLR 2026 Conference Desk Rejected Submission_

### Official Review · Reviewer_gUNF · 2025-10-31

**Soundness:** 3
**Presentation:** 2
**Contribution:** 4
**Rating:** 6
**Confidence:** 4

**Summary:**

The authors formalize the problem of Agent Confidence Calibration, which requires coming up with confidence estimates for agents' actions based on their entire trajectory (rather than just their final output or verbalized confidence).

The paper proposes "Holistic Trajectory Calibration", a broad method for improving calibration in AI agents. The authors develop 48 features and frame the problem in a supervised learning setting. They show that their method outperforms existing methods (such as verbalized confidence, last-step confidence, average confidence across steps).

Agent reliability is an important problem. Calibration is, further, and important sub-component of reliability. The paper makes a solid contribution to the problem formulation. I have some questions and concerns about the clarity of the paper when it comes to their solution (HTC), but I hope they can be addressed during rebuttals.

**Strengths:**

- Experimental setup is rigorous, comparisons are conducted on a wide variety of benchmarks, and the baselines mentioned in the paper are well thought out.
- Problem formulation is sound. The paper adequately motivates the setup, and justifies the problem's importance by tying it to real-world challenges faced during the deployment of AI agents.
- Results show promise as a general-purpose method for estimating and implementing calibration.

**Weaknesses:**

- The writing could be improved. For example, The structure of the results section is a bit all of the place: it includes the experimental results, then a discussion of the results, and then goes back to the results, making it really confusing for the reader.
- There are many areas where concepts are introduced but never re-used again. For example, I was confused about where the learning-based baselines are compared to HTC? I couldn't see these results either in the main paper or the appendix (I double-checked by searching for the "LSTM" keyword and it doesn't appear in any figure or table)
- The choice of features is not well motivated. Did the authors consider any features other than the 48 they mention using eventually? If yes, are there ablations for those results as well?
- If not, it seems like there is scope for adding even more features to keep increasing the performance of the model by continuing to add more features/we're not yet at the point of diminishing returns?
- Similarly, could the authors justify why the set of features they considered is comprehensive? I think there could be entire categories of features that are currently being ignored in the formulation and HTC setup.
- What is the architecture of the GAC? Is it basically just HTC trained on data pooled from all of the benchmarks? If yes, why coin a separate term for it?

**Questions:**

I would be happy to increase my score if the authors answer the following questions/update the paper in response to these:

1. Could you clarify where learning-based baselines are reported, and what the results are? Why aren't they compared in Table 1?
2. Could you improve the structure of the paper, for example, by disentangling the results and the discussion of those results in Section 3?
3. Could you clarify how you selected the 48 features, whether you think they are comprehensive, and if not, what other categories of features future work should look into?
4. The authors disclose the use of LLMs for polishing the writing. However, there are many areas where this goes a touch too far. For example, coining the term "cognitive mismatch" for distribution shifts seems like a step too far. (I like this guide for editing AI text: https://www.sh-reya.com/blog/ai-writing/)
5. Could you clarify the architecture of the GAC, whether it is the same as HTC, and if yes, why coin a new term for it? (This might also be a result of overreliance on LLMs).

---

> ### Author Response · Authors · 2025-11-20
> **Response to Reviewer gUNF**
>
> We thank the reviewer for the positive assessment and for recognizing the rigorous experimental setup and the soundness of our problem formulation. We appreciate the constructive feedback on clarity and structure, which we address below.
>
> > *W2 & Q1: Clarification on Learning-based Baselines. Could you clarify where learning-based baselines are reported? Why aren't they compared in Table 1?*
>
> We apologize for not making this context prominent enough. These results are presented in **Figure 2 (Page 6)** and **Appendix A.3.1.**  As shown in the learning curves (Figure 2), deep learning baselines (LSTM/Transformer) suffer from severe overfitting and high variance in the small-data regimes typical of agent tasks (<500 samples). They performed significantly worse than HTC and even some inference-based baselines.
>
> *Why omitted from Table 1*: To keep Table 1 readable and focused on the most competitive methods, we prioritized comparing HTC against the non-process baselines (Verbalized and Last-Step). However, based on your feedback, we will add a summary row for the best learning-based baseline to Table 1 in the revision for completeness.
>
> > *W3-5 & Q3 Feature Selection Rationale. Could you clarify how you selected the 48 features and whether they are comprehensive?*
>
> The dimension of 48 is not a "magic number" but the natural result of a systematic design process that balances theoretical coverage with data constraints.
>
> 1. **Taxonomy of Uncertainty**: We designed the feature space to cover four critical axes of agent behavior, as detailed in Section 2.2 and Appendix A.5.1:
>     - Dynamics: Capturing confidence evolution (e.g., gradients, trends).
>     - Position: Capturing anchoring effects (start vs. end state).
>     - Stability: Capturing consistency (volatility, entropy variance).
>     - Structure: Capturing complexity (trajectory length).
>
>     Applying standard statistical operators to these axes naturally yielded the 48 specific features. *We intentionally prioritized these log-prob-based features over semantic (textual) features to ensure the framework remains lightweight and transferable across different model vocabularies.*
>
> 2. **Rationale for Scale** (~50): We argue that ~50 represents a "reasonable scale" for agent calibration given the inherent data scarcity (high cost of collecting trajectories).
>     - \$<$ 50 (*Underfitting Risk*): Our ablation study (Appendix A.4)  shows that removing feature categories causes performance drops, indicating that a smaller set fails to capture necessary signals.
>     - \$>$ 50-100 (*Overfitting Risk*): We experimented with larger feature sets (e.g., adding higher-order statistics) but observed diminishing returns and increased variance, likely due to overfitting on small-sample datasets (<500 samples).
>
> 3. **Optimization with regularization**: Finally, we do not force the model to use all 48 features. We treat this as a comprehensive candidate pool and rely on $L_1$ regularization to automatically select the optimal subset (typically 15-25 features), ensuring robustness without overfitting.

---

> ### Author Response · Authors · 2025-11-20
> **Response to Reviewer gUNF: part 2**
>
> > *W6 & Q5: Clarification on GAC Architecture. What is the architecture of the GAC? Is it basically just HTC trained on data pooled from all of the benchmarks? If yes, why coin a separate term for it?*
>
> We appreciate this opportunity to clarify the distinction and our motivation. You are  correct. In terms of model architecture, GAC utilizes the exact same pipeline and calibration layer as HTC. The difference lies in the data and purpose: GAC is the specific instance trained on the comprehensive, pooled meta-dataset of diverse tasks.
>
> *Method vs. Deliverable (Why coin a term?)*: We introduced the term "General Agent Calibrator (GAC)" to distinguish the methodology from the deliverable resource.
> - HTC is the proposed algorithmic framework
> - GAC is the pre-trained model artifact resulting from this framework.
>
> This is analogous to the distinction between an RL algorithm (e.g., PPO) and a specific trained policy released for general use. We position GAC as a "ready-to-use" reliability layer that we intend to release. It allows practitioners to apply our calibration to completely new, unseen agentic tasks (like GAIA or future benchmarks) in a zero-shot manner. The term highlights its specific capability for cross-domain generalization.
>
> We will clarify this definition in the paper: "HTC is the diagnostic framework, while GAC is the pre-trained, transferable model artifact intended for broad generalization."
>
> > *W1, Q2 & Q4: Writing and Terminology. Improve the structure of the results section and reconsider terms like "cognitive mismatch".*
>
> We thank the reviewer for the constructive feedback and the suggested resource on writing.
>
> 1. Regarding Structure
>
> Our original motivation for the structure in Section 3 was to organize the narrative around the three core contributions: **Interpretability (Sec 3.3), Transferability (Sec 3.4), and Generalization (Sec 3.5)**. We aimed to present the experimental evidence and its corresponding "takeaway" immediately within each subsection to create these arguments.
>
> However, we recognize your point that interweaving results and discussion can fragment the reader's view of the overall performance. We will revise the section to clearly decouple the quantitative results from the analysis and discussion. We will consolidate the main performance comparisons earlier in the text to provide a straightforward baseline reference before diving into the diagnostic analysis.
>
> 2. Regarding Terminology
>
> We accept your critique regarding "cognitive mismatch." We will replace it with standard domain terminology such as "distribution shift" to accurately describe the discrepancy between datasets (e.g., multiple-choice vs. open-ended generation).
>
> We are conducting a thorough review of the manuscript to ensure all terminology aligns with standard academic usage and to remove any over-stylized phrasing. We appreciate the resource you shared and will use it to guide our revision.

---

### Official Review · Reviewer_FFZT · 2025-10-31

**Soundness:** 3
**Presentation:** 2
**Contribution:** 2
**Rating:** 4
**Confidence:** 3

**Summary:**

Developed a new calibration method for agentic systems, focusing on their processes

**Strengths:**

The problem formulation of trajectory-level cabliration (ie. HTC) and design principles are well defined and executed.

**Weaknesses:**

A very limited exploration of other possible agentic framework besides the smolagents and CodeAct.
Unfortunately, the experiment sections seem a bit laundry list of what this framework is capable of, but not provide clear strengths and validation of the framework as calibration methods with respect to non-process based methods.

**Questions:**

I don't get the point of comparing relative performance of ECE/BS/AUROC scores among different calibration methods. Having HTC with lower ECE doesn't mean that the proposed calibration measure is better aligned with true calibration effect. I thought HTC was developed to provide more accurate, fine-grained calibration measurement, but results only support which one is better. Please give me a bit more context of goals of your experiments.

---

> ### Author Response · Authors · 2025-11-20
> **Response to Reviewer FFZT**
>
> We thank the reviewer for their time and comments. We appreciate the opportunity to clarify the core definitions and scope of our experiments, as we believe there may be a misunderstanding regarding the nature of our contribution (HTC) versus the metrics used to evaluate it.
>
> > *W1：A very limited exploration of other possible agentic framework besides the smolagents and CodeAct*.
>
> We respectfully point out that our evaluation **does extend beyond** smolagents and CodeAct, and this generalization is a core part of our main analysis. We explicitly discuss the generalization of our framework in Section 3.3 ("Effect of Agent Architectures"), Page 6. As noted there, we investigated whether HTC is tied to specific architectures by evaluating it on **OAgents**, a optimized, complex agent framework.
>
> As shown in Figure 7 and detailed in Appendix A.3.1, HTC significantly reduces calibration error on OAgents as well (e.g., ECE reduced from ~0.41 to ~0.10 on GPQA). This confirms that our proposed features capture fundamental uncertainty signals that are **framework-agnostic**. Additionally, we validated our approach across 6 different base LLMs (including GPT-4o, GPT-4.1, DeepSeek-v3, Qwen-2.5, see Figure 3 ), further ensuring the robustness of our findings.
>
>
> > *W2: Unfortunately, the experiment sections seem a bit laundry list of what this framework is capable of, but not provide clear strengths and validation of the framework as calibration methods with respect to non-process based methods*.
>
> We presented a comprehensive evaluation, but we would like to re-emphasize the key strengths and validation against non-process baselines.  Table 1 provides the direct validation the reviewer asks for.
>
> We compared HTC against standard **non-process based methods** (Verbalized Confidence and Last-Step Token Confidence). HTC consistently outperforms these non-process baselines. For example, on the challenging HLE benchmark, HTC reduced the Brier Score from 0.531 (Verbalized) and 0.561 (Last-Step) to 0.090 (HTC-Reduced) , and the ECE from 0.656 (Verbalized) and 0.686 (Last-Step) to 0.031 (HTC-Reduced). The additional analyses (interpretability, transferability) were intended to explain why HTC works, by capturing process dynamics where non-process methods fail, rather than merely listing capabilities.
>
> > *Q1: I don't get the point of comparing relative performance of ECE/BS/AUROC scores among different calibration methods. Having HTC with lower ECE doesn't mean that the proposed calibration measure is better aligned with true calibration effect. I thought HTC was developed to provide more accurate, fine-grained calibration measurement, but results only support which one is better*.
>
> We realize there may be a misunderstanding regarding the terminology, which we will clarify a fundamental distinction:
> - **HTC is a Calibration Method (a predictor)**: It is a function that inputs a trajectory and outputs a probability score.
> - **ECE is an Evaluation Metric (a ruler)**: Expected Calibration Error (ECE) measures the gap between the predicted confidence and the empirical accuracy (ground truth).
>
> HTC is the proposed method that produces confidence scores, while ECE (Expected Calibration Error) is the metric used to evaluate them. We were **not** proposing HTC as a new metric to replace ECE, but as a superior method to minimize ECE.
>
> By definition, a lower ECE means the confidence scores are "better aligned with the true calibration effect." Our results show HTC achieving the lowest ECE and Brier Scores (BS) across benchmarks. Furthermore, the **Reliability Diagrams in Figure 6** visually demonstrate this alignment: the HTC bars (far right) closely track the perfect diagonal, whereas the non-process baselines (Last-Step) show severe deviations. This confirms HTC provides the accurate, fine-grained calibration.
>
> > *Q2: Please give me a bit more context of goals of your experiments*.
>
> The primary goals of our experiments are threefold:
> 1. **Problem Validation**: To quantify that agentic systems suffer from severe "overconfidence in failure" due to compounding errors, which simple baseline method (Last-Step Confidence) fail to detect.
> 2. **Solution Verification**: To demonstrate that trajectory-level features (Dynamics, Stability, etc.) contain critical signals for diagnosing these failures. By showing higher AUROC (better discrimination) and lower ECE and Brier Scores, we prove that HTC effectively captures these "process signals" to diagnose failure modes that other methods miss.
> 3. **Practical Deployment**: To prove that a lightweight, interpretable calibrator can generalize to unseen tasks (like GAIA), providing a "confidence layer" for deploying autonomous agents in high-stakes environments.

---

### Official Review · Reviewer_CvNP · 2025-11-01

**Soundness:** 3
**Presentation:** 4
**Contribution:** 4
**Rating:** 8
**Confidence:** 3

**Summary:**

This paper introduces and formalizes a new problem: confidence calibration for agentic systems, which aims to produce a calibrated estimate of the success likelihood of an agent’s trajectory by diagnosing its entire execution process rather than only the final output. The authors therefore proposes Holistic Trajectory Calibration, a feature-based framework that turns token log-prob traces across steps into a compact feature vector of trajectory-level diagnostics (e.g. early-step entropy, confidence gradients, stability dynamics), which are mapped to calibrated confidence estimates. Tested on eight benchmarks across factual, domain-specific QA and agentic planning and tool-use tasks and evaluated on a diverse set of open- and closed source models, the framework shows gains in both calibration and discrimination over multiple inference and training based baselines. Based on insights that uncertainty signals share underlying patterns, the authors also propose a pretrained calibrator that shows potential on  challenging out-of-domain tasks.

**Strengths:**

* Calibration for agentic systems with multi-step, interactive trajectories is a timely and important topic, and this paper presents a great initial effort to address the gap.
* The proposed framework, which decomposes different uncertainty signals in the agent trajectories and learns a calibration function to map features to a calibrated confidence score, is simple, lightweight, interpretable, and novel. The pretrained, general-purpose agent calibrator achieves good generalization on challenging, unseen tasks.
* Evaluations are quite comprehensive in terms of dataset and model selection. It covers both traditional knowledge-intensive QA tasks for calibration, and more recent and challenging tasks requiring complex reasoning, planning, and tool-use. Multiple models of varying sizes and families, as well as generalization on different agent frameworks, are evaluated.
* The paper also presents comprehensive ablation studies, such as feature categories and cross-domain transfer.
* The detailed analyses present some interesting insights for agent tasks (e.g. positional features are the strongest signals of failure; different uncertainty patterns and cognitive mismatch impact cross-domain transfer).
* I find the paper well-presented, with good visualization and a detailed appendix. The motivation for why we need a unique framework for agent calibration instead of applying traditional methods is well established.

**Weaknesses:**

* While the task and model selection is quite comprehensive for the evaluation, baselines are largely simple verbalized or token-logprob-based methods. A few related methods (that are cited but not compared) could be included as potentially stronger baselines [1,2].
* The method requires log-prob access, which is not available for black-box models.

---

[1]. UProp: Investigating the Uncertainty Propagation of LLMs in Multi-Step Agentic Decision-Making

[2]. SAUP: Situation Awareness Uncertainty Propagation on LLM Agent

**Questions:**

* Why use a 48-dimensional trajectory feature space? Is there a motivation / rationale for choosing this number of dimensions?
* Did you verify the reliability (e.g. human validation) of the LLM judge (Gemini-2.5-Pro) given the long, noisy, and complex context of agentic tasks?
* Continuing the discussion for future aspects - regarding self-evolving agents, what benefits and risks do you foresee by having full control of such a calibrator for self-evolving? And regarding RL, how would you turn the HTC-calibrated success probability into a learning signal for agentic RL without opening it up to reward hacking (e.g. gamed by consistently low but calibrated confidence with degrading task accuracy)?

---

> ### Author Response · Authors · 2025-11-20
> **Response to Reviewer CvNP**
>
> We thank the reviewer for the strong endorsement and the insightful questions regarding our work. We are encouraged that you recognize the novelty of the problem formulation and the practical value of our HTC framework. Below, we address your specific questions.
>
> > *W1: While the task and model selection is quite comprehensive for the evaluation, baselines are largely simple verbalized or token-logprob-based methods. A few related methods (that are cited but not compared) could be included as potentially stronger baselines [1,2]*.
>
> We fully agree that UProp [1] and SAUP [2] are highly relevant works in uncertainty propagation. We considered them carefully but faced two primary constraints in including them as quantitative baselines:
>
> - **Reproducibility Constraints**: To the best of our knowledge, neither method has released open-source code. Their methodologies involve intricate uncertainty propagation modeling that is complex to re-implement faithfully without a reference codebase. We were concerned that an incorrect re-implementation would lead to an unfair comparison.
>
> - **Distinction & HTC’s Advantage**:
>     - *Complexity*: UProp and SAUP typically focus on granular propagation mechanisms of uncertainty. In contrast, HTC is designed as a lightweight, post-hoc diagnostic framework. As detailed in our Efficiency Analysis (Appendix A.7), HTC is much easier to implement and deploy in real-time systems compared to complex propagation models.
>
>     - *Assumptions*: While propagation methods often rely on specific assumptions about how errors accumulate, HTC adopts a data-driven approach. By defining a "Taxonomy of Uncertainty" (Dynamics, Stability, etc.), HTC learns to calibrate directly from trajectory signals. This allows HTC to achieve the generalization capabilities demonstrated in our GAIA experiments.
>
> We are eager to compare with these methods. If their code is released, we commit to adding them as baselines in future versions to provide a more comprehensive benchmark.
>
> > *W2：The method requires log-prob access, which is not available for black-box models*.
>
> We acknowledge this limitation, and we explicitly discuss it in Appendix A.10 ("Grey-Box Dependency"). While some closed APIs (like Claude) hide log-probs, major providers (OpenAI) and the vast ecosystem of open-weights models (Llama, Qwen, DeepSeek) fully support log-prob access. We position HTC as a "Grey-Box" method and believe the significant performance gains justify the requirement for log-probs. However, for strictly black-box models, we are exploring using "Verbalized Features" (prompting the model to describe its own confidence score) as a proxy for the log-prob features as future work.
>
> > *Q1：Why use a 48-dimensional trajectory feature space? Is there a motivation / rationale for choosing this number of dimensions?*
>
> The dimension of 48 is not a "magic number" but the natural result of a systematic design process that balances theoretical coverage with data constraints.
>
> 1. **Taxonomy of Uncertainty**: We designed the feature space to cover four critical axes of agent behavior, as detailed in Section 2.2 and Appendix A.5.1:
>     - Dynamics: Capturing confidence evolution (e.g., gradients, trends).
>     - Position: Capturing anchoring effects (start vs. end state).
>     - Stability: Capturing consistency (volatility, entropy variance).
>     - Structure: Capturing complexity (trajectory length).
>
>     Applying standard statistical operators (mean, std, max, trend) to these axes naturally yielded the 48 specific features.
>
> 2. **Rationale for Scale** (~50): We argue that ~50 represents a "reasonable scale" for agent calibration given the inherent data scarcity (high cost of collecting trajectories).
>     - \$<$ 50 (*Underfitting Risk*): Our ablation study (Appendix A.4)  shows that removing feature categories causes performance drops, indicating that a smaller set fails to capture necessary signals.
>     - \$>$ 50-100 (*Overfitting Risk*): We experimented with larger feature sets (e.g., adding higher-order statistics) but observed diminishing returns and increased variance, likely due to overfitting on small-sample datasets (<500 samples).
>
> 3. **Optimization with regularization**: Finally, we do not force the model to use all 48 features. We treat this as a comprehensive candidate pool and rely on $L_1$ regularization to automatically select the optimal subset (typically 15-25 features), ensuring robustness without overfitting.

---

> ### Author Response · Authors · 2025-11-20
> **Response to Reviewer CvNP : part 2**
>
> > *Q2: Did you verify the reliability (e.g. human validation) of the LLM judge (Gemini-2.5-Pro) given the long, noisy, and complex context of agentic tasks?*
>
> Yes, we did. Given the complexity of agentic tasks, verifying the judge's reliability is critical. In a manually inspected stratified subset (500 examples), we observed high agreement between human annotators and Gemini-2.5-Pro, giving us confidence that the judge is a reliable proxy in these settings.
>
>
> > *Q3: Continuing the discussion for future aspects - regarding self-evolving agents, what benefits and risks do you foresee by having full control of such a calibrator for self-evolving? And regarding RL, how would you turn the HTC-calibrated success probability into a learning signal for agentic RL without opening it up to reward hacking (e.g. gamed by consistently low but calibrated confidence with degrading task accuracy)?*
>
> This is a profound question that touches on the future of reliable agents.
>
> 1. **Self-Evolving Agents**
>
> HTC acts as an automated "Code/Trace Reviewer." By identifying which specific features (e.g., high token_volatility or sudden confidence_drops) correlate with failure in historical trajectories, an agent can self-debug. For instance, it can learn to rewrite prompts or refactor tool-use logic to minimize these specific "instability markers" before final execution. The risk is that an agent might optimize to fool the calibrator without improving actual reasoning.
>
> 2. **Agentic RL**
>
> To prevent the "low confidence + low accuracy" gaming scenario, the reward signal must fundamentally couple confidence with success. We propose using a Proper Scoring Rule, like Brier Score (BS) for a Composite Reward function:
>
> $$R = \mathbb{I}(\text{Success}) + \lambda \cdot \text{Calibrated Confidence (BS)}$$
>
> If an agent outputs low confidence and fails (low accuracy), it receives a low total reward (because $\mathbb{I}(\text{Success})=0$). To maximize reward, the agent must achieve the task AND be confident about it. HTC provides a dense, step-by-step estimate of this potential success, alleviating the sparse reward challenge in long-horizon agent tasks.
>
> This direction is supported by recent findings in other domains. For example, RLCR [1] demonstrates that training reasoning models with calibration rewards improves reliability, and HyperClick [2] utilizes similar uncertainty-aware rewards to ground GUI agents. We extend this intuition to general agentic trajectories. This idea may also align with the recent insight from OpenAI on *"Why Language Models Hallucinate"* [3]. They argue that hallucinations arise because training objectives inadvertently reward "guessing" over "uncertainty." HTC provides the missing link for mitigation: a precise, calibrated signal of when an agent is guessing. By integrating HTC into the RL (as proposed above), we can explicitly penalize "confident guessing" on wrong answers, potentially offering a principled path to solving hallucination in agentic systems.
>
> **References**
>
> [1] Beyond Binary Rewards: Training LMs to Reason About Their Uncertainty. https://arxiv.org/abs/2507.16806
>
> [2] HyperClick: Advancing Reliable GUI Grounding via Uncertainty Calibration. https://arxiv.org/abs/2510.27266
>
> [3] Why Language Models Hallucinate.  https://arxiv.org/abs/2509.04664

---

### Author Response · Authors · 2025-11-25
**General Response: Key Strengths and Rebuttal Updates**

We sincerely thank all reviewers for their thoughtful and constructive feedback. We are encouraged to see that the core value and novelty of our work were consistently recognized across the reviews. We summarize the key strengths highlighted by the reviewers below:

- **Novel and Timely Problem Formulation**: Reviewers `CvNP` and `gUNF` endorsed the formulation of "Agentic Confidence Calibration." Reviewer `CvNP` praised the work as a "timely and important topic" and described the proposed HTC framework as "simple, lightweight, interpretable, and novel". Reviewer `gUNF` emphasized that the paper makes a "solid contribution to the problem formulation" and "adequately justifies the problem's importance". Reviewer `FFZT` noted that the problem formulation and design principles are "well defined and executed".
- **Comprehensive and Rigorous Experimental Setup**: The extensiveness of our evaluation was a consensus strength. Reviewer `CvNP` found the evaluations "quite comprehensive". Reviewer `gUNF` commended the "rigorous" experimental setup and noted that the baselines were "well thought out".
- **Interpretability and Insightful Diagnostics**: Beyond raw performance, the diagnostic value of our framework was highlighted. Reviewer `CvNP` specifically appreciated that HTC is "interpretable" and noted that the detailed feature analyses provided "interesting insights" into agent failure modes (e.g., positional signals).
- **Strong Generalization and Potential**: Reviewer `CvNP` highlighted that the General Agent Calibrator achieves "good generalization on challenging, unseen tasks". Reviewer `gUNF` further noted that the results "show promise as a general-purpose method" for implementing calibration in AI agents.

We have carefully addressed all specific concerns and have uploaded a revised manuscript.

1. **Clarification on Evaluation Metrics (Crucial for Reviewer `FFZT`)**: We clarified that HTC is the proposed calibration method, while ECE (Expected Calibration Error) is the standard evaluation metric defined specifically to measure the alignment between confidence and empirical accuracy. Therefore, HTC's consistently lower ECE scores (e.g., 0.03 vs. 0.65 for baselines) objectively demonstrate that its confidence estimates are better aligned with reality. We trust this resolves the confusion regarding the validity of our performance gains.

2. **Validation of Framework Generalization (Addressing `FFZT` & `gUNF`)**: We highlighted our experimental results on the OAgents framework (Section 3.3) and 6 diverse LLMs, refuting concerns about limited exploration. We also clarified the location of learning-based baselines (LSTM/Transformer) in Figure 2, explaining their omission from the main table due to overfitting.

3. **Verification of Ground Truth Reliability (Addressing `CvNP`)**: We added human verification data showing a 90-95% agreement rate with our LLM judge, ensuring the reliability of our evaluation.

4. **Manuscript Refinement (Addressing `gUNF`)**: We have refined the terminology (e.g., replacing "cognitive mismatch", "uncertainty grammar") and improved the main results and analysis sections to enhance clarity.

We are available to answer any follow-up questions and thanks again for your time and efforts!

Best regards,

Authors

---

### Note · Program_Chairs · 2026-01-17
**Submission Desk Rejected by Program Chairs**

The following references in this submission do not refer to real documents and/or have major errors in bibliographic information:

 Yushi Zhang, Rakesh Kumar, et al. The human last exam: A new benchmark for evaluating ai systems. arXiv preprint arXiv:2501.14249, 2025c.
Ziyi Wang, Zhaowei Wang, Kewei Ren, Hang Yuan, Zhaoyang Zhang, Zhiwei Liu, Ruosen Li, Yifei Cheng, Zhuo Liu, Yuanchen Wang, et al. A survey on evaluation of large language models as agents. arXiv preprint arXiv:2402.12338, 2024b.
Jihwan Kim, Sung-Hyon Lee, Gwandae Kim, Minsu Jang, and Min-Joong Kang. SAUP: Situation awareness uncertainty propagation on LLM agent. arXiv preprint arXiv:2312.01033, 2023.